# Repeatome Analyses and Satellite DNA Chromosome Patterns in *Deschampsia sukatschewii*, *D. cespitosa*, and *D. antarctica* (Poaceae)

**DOI:** 10.3390/genes13050762

**Published:** 2022-04-26

**Authors:** Alexandra V. Amosova, Olga Yu. Yurkevich, Nadezhda L. Bolsheva, Tatiana E. Samatadze, Svyatoslav A. Zoshchuk, Olga V. Muravenko

**Affiliations:** Engelhardt Institute of Molecular Biology, Russian Academy of Sciences, 32 Vavilov St., 119991 Moscow, Russia; olikys@gmail.com (O.Y.Y.); nlbolsheva@mail.ru (N.L.B.); tsamatadze@gmail.com (T.E.S.); slavazo@mail.ru (S.A.Z.); olgmur1@yandex.ru (O.V.M.)

**Keywords:** high-throughput sequencing, *Deschampsia sukatschewii*, *D. antarctica*, *D. cespitosa*, repeatome, chromosome, FISH

## Abstract

Subpolar and polar ecotypes of *Deschampsia sukatschewii* (Popl.) Roshev, *D. cespitosa* (L.) P. Beauv, and *D. antarctica* E. Desv. are well adapted to stressful environmental conditions, which make them useful model plants for genetic research and breeding. For the first time, the comparative repeatome analyses of subpolar and polar *D. sukatschewii*, *D. cespitosa*, and *D. antarctica* was performed using RepeatExplorer/TAREAN pipelines and FISH-based chromosomal mapping of the identified satellite DNA families (satDNAs). In the studied species, mobile genetic elements of class 1 made up the majority of their repetitive DNA; interspecific variations in the total amount of Ty3/Gypsy and Ty1/Copia retroelements, DNA transposons, ribosomal, and satellite DNA were revealed; 12–18 high confident and 7–9 low confident putative satDNAs were identified. According to BLAST, most *D. sukatschewii* satDNAs demonstrated sequence similarity with satDNAs of *D. antarctica* and *D. cespitosa* indicating their common origin. Chromosomal mapping of 45S rDNA, 5S rDNA, and satDNAs of *D. sukatschewii* allowed us to construct the species karyograms and detect new molecular chromosome markers important for *Deschampsia* species. Our findings confirmed that genomes of *D. sukatschewii* and *D. cespitosa* were more closely related compared to *D. antarctica* according to repeatome composition and patterns of satDNA chromosomal distribution.

## 1. Introduction

Several species of the cosmopolitan grass genus *Deschampsia* P. Beauv. (Poaceae) are well adapted to stressful environmental conditions including extreme polar habitats [1,2,3]. In particular, polar and subpolar ecotypes of *D. sukatschewii* (Popl.) Roshev and *D. cespitosa* (L.) P. Beauv. are widespread in the Arctic and sub-Arctic regions of Canada, Europe, Siberia, Chukotka Peninsula, and the Altai mountains [1,4,5,6]. *D. antarctica* E. Desv. is one of two native angiosperms adapted to extreme Antarctic environments, which can be found in diverse Antarctic habitats, including the west coast of the Antarctic Peninsula, the Maritime Antarctic, sub-Antarctic Islands, and northern Patagonia [3,7,8,9]. Such native cold-hardy ecotypes of the *Deschampsia* species are resources of genes associated with environmental stress tolerance and can also serve as models in crop breeding strategies [10,11].

Plant responses to environmental stresses might include some genetic changes (e.g., alternation in metabolic pathways and transcriptional regulation of genes) and cytological alterations [12]. Currently, genome diversity and comparative chromosomal phylogeny of cold-hardy ecotypes of *Deschampsia* are being intensively studied. Transcriptome sequencing of *D. antarctica* has been performed under various abiotic stress conditions and its expression profile has been examined [10]. For *D. antarctica* populations from the Maritime Antarctic, retrotransposon-based genetic polymorphism was reported, which could be related to the environmentally induced mobilization of random transposable elements as well as unique reproductive features of this species [13]. For polar and subpolar ecotypes of *Deschampsia*, structural chromosomal variations, including chromosome rearrangements, aneusomaty, mixo- and aneuploidy were revealed [14,15,16], indicating importance of molecular cytogenetic characterization of such plants. A comparative molecular cytogenetic analysis of several *Deschampsia* species, including subpolar ecotypes of *D. antarctica*, *D. cespitosa*, and *D. sukatschewii*, was performed using fluorescence *in situ* hybridization (FISH) with 45S/5S rDNA and sequential rapid genomic *in situ* hybridization with genomic DNAs of closely related and distant species of this genus. As a result, some intra- and interspecific differences in their karyotypes were revealed [16].

Repetitive DNAs (including mobile genetic elements and tandem repetitive DNA (satellite DNA)) are the major and fast-evolving part of genomes of vascular plants, which could contribute to speciation processes [13,17,18,19,20,21]; and comparative repeatome analyses make it possible to identify genomic differences both in closely related and distant plant taxa. Several satellite DNA families (satDNAs), including CON/COM repeats, are shared between many taxa of the Aveneae/Poeae tribe, indicating that they can be used as effective molecular and chromosomal markers for characterizing the plant genome, for assessing intra- and interspecific variability of genomes, and also in phylogenetic studies [22,23,24]. Recently, CON/COM satDNAs were identified and characterized in genomes of several species of *Deschampsia* and related genera, which made it possible to clarify some phylogenetic relationships between these species [25].

One of the effective modern approaches for characterization of repetitive DNA in one or several plant species includes genome-wide bioinformatic analyses by RepeatExplorer/TAREAN (Tandem Repeat Analyzer) pipelines, which use graph-based clustering and analyze next-generation sequencing data [26,27,28]. Having many advantages (e.g., it does not require a reference genome for contigs assembling, offers an easy-to-use interface, a rather fast analysis with detailed results), these pipelines are frequently used to create repeat databases and also to identify satDNAs suitable as FISH probes for further molecular cytogenetics [29,30,31,32,33]. Recently, a number of satDNAs were identified in genomes of South American accessions of *D. antarctica* and *D. cespitosa*, and also chromosomal distribution of some satDNAs were analyzed in several *Deschampsia* species growing in the same region [34,35]. However, for a thorough understanding of the relationship between species within the genus *Deschampsia*, further studies of the genomic diversity are needed. In particular, intra- and interspecific variability in the composition and genomic organization of transposable elements, as well as satDNA should be explored in different *Deschampsia* species and accessions from other growing areas.

The present work performed a comparative characterization of repeatomes of subpolar and polar accessions of *D. sukatschewii*, *D. cespitosa*, and *D. antarctica*, including genome-wide bioinformatic analyses of low-coverage high-throughput DNA sequencing data using RepeatExplorer and TAREAN pipelines and the Basic Local Alignment Search Tool (BLAST). Also, FISH-based chromosomal mapping of the identified specific satDNAs and a search for new molecular chromosome markers were carried out to provide information about the changes that might occur in their genomes during speciation.

## 2. Materials and Methods

### 2.1. Plant Material

Seeds of *D. antarctica* (KEW-0661919, Falkland Is., UK) were obtained from the collection of Weddle Seed Conservation Department, Royal Botanic Gardens, Kew, UK. Seeds of *D. cespitosa* (688, Vologda region, Russia) and *D. sukatschewii* (78, Altai Mountains, Russia) were provided by the laboratory of genetic resources of fodder plants, Federal Williams Research Center of Forage Production and Agroecology (FWRC FPA), Moscow, Russia. Seeds of the studied accessions were germinated in Petri dishes on the moist filter paper for 3–5 days. Then the plants were grown in a greenhouse at 15 °C.

### 2.2. Genomic DNA Extraction and Sequencing

Genomic DNA of *D. sukatschewii* and *D. cespitosa* were isolated from young leaves of the studied accessions using the GeneJet Plant Genomic DNA Purification Kit (Thermo Fisher Scientific, Vilnius, Lithuania). The quality of the DNA samples was checked with the Implen Nano Photometer N50 (Implen, Munich, Germany). The concentration and purification of the extracted DNAs were assessed with the Qubit 4.0 fluorometer and Qubit 1X dsDNA HS Assay Kit (Thermo Fisher Scientific, Eugene, OR, USA).

For *D. sukatschewii* and *D. cespitosa*, whole genome sequencing with low coverage was performed at the Beijing Genomics Institute (BGISeq platform) (Shenzhen, Guangdong, China) according to the NGS protocol for generating 5–10 million of paired-end reads of 150 bp in length, which provided at least 0.147–0.350× of the coverage of the *D. cespitosa* genome (1C = 4283.64–5105.16 Mbp, Eurasian region) [36,37]. The raw sequencing data for *D. cespitosa* (SAMN26938767) and *D. sukatschewii* (SAMN26938768) were uploaded to the National Center for Biotechnology Information (NCBI) BioProject database under accession number PRJNA819861 (https://www.ncbi.nlm.nih.gov/bioproject/PRJNA819861, accessed on 25 March 2022).

### 2.3. Sequence Analysis and Identification of DNA Repeats

For genome-wide comparative analyses, genome sequences of *D. sukatschewii* and *D. cespitosa* and also the publicly available *D. antarctica* sequencing data (https://www.ebi.ac.uk/ena/browser/view/PRJNA237267?show=reads, (accessed on 28 January 2018) PRJNA237267 gDNA-Seq for Antarctic hairgrass, Korea Polar Research Institute), were used. Interspecific comparisons, reconstruction, and quantification of major repeat families were performed with the use of RepeatExplorer 2 and TAREAN pipelines [27,38]. For each studied species, the genomic reads were filtered by quality, and 1 million high-quality reads were randomly selected for further analyses, which corresponds to 0.0147–0.0350× of a coverage of the *D. cespitosa* genome (1C = 4283.64–5105.16 Mbp, Eurasian region) [36,37], and is within the limits recommended by the developers of these programs (genome coverage of 0.01–0.50× is recommended) [38]. RepeatExplorer/TAREAN was launched with the preset settings based on Galaxy platform (https://repeatexplorer-elixir.cerit-sc.cz/galaxy/, 25 March 2022). Initially, the preprocessing of the genomic reads was performed. The reads were filtered in terms of quality using a cut-off of 10, trimmed, and filtered by size to obtain high-quality reads. Default threshold was explicitly set to 90% sequence similarity spanning at least 55% of the read length (in the case of reads differing in length it applies to the longer one). The sequence homology of the identified satDNAs of *D. sukatschewii* with repeats of *D. cespitosa* and *D. antarctica* was estimated by BLAST (NCBI, Bethesda, MD, USA). Based on twelve abundant satDNAs of *D. sukatschewii*, oligonucleotide FISH probes Ds 52, Ds 56, Ds 65, Ds 81, Ds 83, Ds 88, Ds 124, Ds 138, Ds 144, Ds 146, Ds 179, and Ds 226 (Table 1) were generated by the Primer3-Plus software [39].

### 2.4. Chromosome Spread Preparation

Root tips (0.5–1 cm long) were kept in ice water for 24 h for accumulation of mitotic divisions and then fixed in the ethanol and glacial acetic acid fixative (3:1) for 2 days at room temperature. The fixed roots were incubated in 1% acetocarmine solution (in 45% acetic acid) for 30–40 min. Then, the root meristem was cut from the tip cap, macerated in 45% acetic acid, and a squashed preparation was made with the use of a cover slip. After freezing in liquid nitrogen, the cover slip was removed; the obtained preparation was dehydrated in 96% ethanol for 3 min and air dried for 15 min.

### 2.5. Fluorescence In Situ Hybridization

In FISH assays, we used two wheat DNA probes: pTa71 enclosing 18S-5.8S-26S (45S) rDNA [40] and pTa794 containing 5S rDNA [41]. These DNA probes were labeled directly with fluorochromes Aqua 431 dUTP, Red 580 dUTP, or Green 496 dUTP (ENZO Life Sciences, Farmingdale, NY, USA) by nick translation according to manufacturers’ protocols. Moreover, oligonucleotide probes Ds 52, Ds 56, Ds 65, Ds 81, Ds 83, Ds 88, Ds 124, Ds 138, Ds 144, Ds 146, Ds 179, and Ds 226 were used. These probes were produced and labeled directly with 6-FAM- or Cy3-dUTP in *Evrogen JSC* (Moscow, Russia).

Several sequential FISH procedures were performed with various combinations of these labeled DNA probes as described previously [6,42]. Before the first FISH procedure, chromosome slides were pretreated with 1 mg/mL RNase A (Roche Diagnostics, Mannheim, Germany) in 2 × SSC at 37 °C for 1 h. Then, the slides were washed three times for 10 min in 2 × SSC, dehydrated through a graded ethanol series (70%, 85%, and 96%) for 3 min each and air dried for 15 min. A total of 15 µL of hybridization mixture containing 40 ng of each labeled probe was added to each slide. The slides with DNA probes were covered with coverslips, sealed with rubber cement, denatured at 74 °C for 5 min, chilled on ice and placed in a moisture chamber at 37 °C. After overnight hybridization, the slides were washed in 0.1 × SSC (10 min, 44 °C), twice in 2 × SSC for 10 min at 44 °C, followed by a 5-min wash in 2 × SSC and three 3-min washes in PBS at room temperature. Then, the slides were dehydrated through the graded ethanol series for 3 min each, air dried for 15 min, and stained with DAPI (4′,6-diamidino-2-phenylindole) dissolved (0.1 μg/mL) in Vectashield mounting medium (Vector Laboratories, Burlingame, CA, USA). After documenting FISH results, the chromosome slides were washed twice in 2 × SSC for 10 min. Then, sequential FISH procedures were conducted on the same slides.

### 2.6. Chromosome Analysis

The chromosome slides were inspected using the epifluorescence Olympus BX61 microscope with the standard narrow band pass filter set and UPlanSApo 100×/1.40 oil UIS2 objective (Olympus, Tokyo, Japan). Chromosome images were captured with a monochrome CCD (charge-coupled device) camera (Snap, Roper Scientific, Tucson, AZ, USA) in grayscale channels, pseudo-colored, and processed with Adobe Photoshop 10.0 (Adobe Systems, Birmingham, AL, USA) and VideoTesT-FISH 2.1 (IstaVideoTesT, St. Petersburg, Russia) software. At least five plants and 15 metaphase plates were examined in each sample. Chromosome pairs in karyotypes were identified according to the chromosome size and morphology, localization of chromosome markers, and also the cytological nomenclature proposed previously [16].

## 3. Results

### 3.1. Comparative Analyses of the Repetitive DNA Sequences

The comparative repeatome analysis of *D. antarctica*, *D. cespitosa*, and *D. sukatschewii* showed that mobile genetic elements made up the majority of their repetitive DNAs (Table 2). Retrotransposon elements, including Ty3-Gypsy and Ty1-Copia superfamilies (transposable elements of Class I), were highly abundant and represented 41.21–43.41% of their genomes. Within the Ty1-Copia superfamily, SIRE and Angela were most abundant, and Ty3-Gypsy retroelements were dominated by the Tat-Retand and Athila non-chromoviruses and chromovirus Tekay. In *D. cespitosa* and *D. sukatschewii*, Ty3-Gypsy elements significantly exceeded Ty1-Copia retrotransposons. In *D. antarctica*, however, Ty1-Copia retroelements were roughly twice abundant than Ty3-Gypsy elements. The genome of *D. antarctica* contained the largest proportion of unclassified LTR retroelements (14.03%) if compared with *D. cespitosa* (1.31%) and *D. sukatschewii* (4.27%). DNA transposons (Class II) were found in lower amount (2.48–2.89%) compared to retrotransposons, and the least quantity was revealed in *D. sukatschewii*. The total amount of satellite DNA ranged from 1.61% (*D. sukatschewii*) to 2.85% (*D. cespitosa*). The content of ribosomal DNA was notably less in *D. antarctica* (0.06%) if compared with *D. sukatschewii* (0.29%) and *D. cespitosa* (0.6%). In the studied accessions, 12–18 high confident and 7–9 low confident putative satellites were revealed by TAREAN (Figure 1, Table 2).

### 3.2. BLAST Analysis

According to BLAST, most of the satDNAs identified in the genome of *D. sukatschewii* (Ds 52, Ds 56, Ds 81, Ds 83, Ds 88, Ds 124, Ds 138, Ds 142, Ds 166, Ds 179, Ds 182, and Ds 226) demonstrated sequence similarity with the satDNAs of *D. antarctica* and/or *D. cespitosa*, and also the species belonged to other genera including *Festuca*, *Helictotrichon*, *Leymus*, *Poa*, *Secale*, *Setaria*, *Tripidium*, and *Triticum* (Table 3). Four Ds satDNAs (Ds 65, Ds 144, Ds 158, and Ds 211) demonstrated sequence homology only with the satDNAs of *D. antarctica* (Da satDNAs) and/or *D. cespitosa* (Dc satDNAs). For Ds 146 satDNA, homology with tandem repeats of other species was not revealed within available NCBI database (Table 3).

### 3.3. Chromosomal Structural Variations

The performed karyotype analyses showed that the studied *Deschampsia* accessions presented diploid karyotype with 2n = 2x = 26 chromosomes (Figure 2, Figure 3, Figure 4 and Figure 5).

In karyotypes of *D. sukatschewii* and *D. cespitosa*, similar patterns of chromosome distribution of 45S and 5S rDNA clusters were observed. Six bright 45S rDNA signals were detected in the short arms of chromosome pairs 5, 6, and 9 with satellites of different sizes and secondary constrictions. Ten hybridization signals of 5S rDNA were observed on chromosome pairs 1 (in the proximal regions of both arms), 3 (in the terminal regions of the long arms), and also in the proximal regions of the long arms of chromosome pairs 7 and 10 (Figure 4 and Figure 5).

In the karyotype of *D. antarctica*, four hybridization signals of 45S rDNA were revealed in the short arms of two chromosome pairs 5 and 9 with satellites of different sizes and secondary constrictions (Figure 4 and Figure 5). Ten loci of 5S rDNA were localized on chromosome pairs 1 (in the proximal regions of the short arm), 3 (in the terminal regions of the long arms), 6 (in the distal regions of the short arms), and also in the proximal regions of the long arms of chromosome pairs 7 and 10 (Figure 4 and Figure 5).

We observed different patterns of chromosome distribution of twelve Ds satDNAs (Ds 52, Ds 56, Ds 65, Ds 81, Ds 83, Ds 88, Ds 124, Ds 138, Ds 144, Ds 146, Ds 179, and Ds 226) in karyotypes of *D. sukatschewii*, *D. cespitosa*, and *D. antarctica*, which exhibited interspecific differences in their clustered and/or dispersed localization (detailed in Appendix A, Figure 2, Figure 3, Figure 4 and Figure 5).

Based on the distribution patterns of the studied molecular cytogenetic markers, chromosomal rearrangements were detected in some karyotypes of *D. sukatschewii* (t(1; 2) and t(6; 9)), *D. cespitosa* (t(2; 3) and t(3; 4)) (Figure 4), and *D. antarctica* (t(3; 13)) (Figure 5).

## 4. Discussion

Most eukaryotic genomes contain large numbers of repetitive DNA sequences [43,44]. Transposable elements (TEs) as well as tandem repeats (satellite DNA) are highly abundant and diverse parts of genomes [45,46]. In plants, TEs can constitute up to 90% of their genomes [47,48,49]. Due to the fact that TEs are capable of changing their location and/or copy numbers, they can influence the genome organization and evolution [50,51]. Currently, TEs are separated into two major classes, class I (retrotransposons, including LTR retrotransposons) and class II (DNA transposons), based on TEs structural characteristics and mode of replication [50,52]. In plant genomes, LTR retrotransposons include the Ty1-Copia and Ty3-Gypsy superfamilies, which are further divided into a number of families mostly specific to a single or a group of closely related species [53]. In plant genomes, LTR retrotransposons are highly abundant, making up to 75% of nuclear DNA [54,55]. In our study, a comparative repeatome analysis of *D. sukatschewii*, *D. cespitosa*, and *D. antarctica* also showed that LTR retrotransposons made up the majority of their genomes. LTR retrotransposons are considered to be main contributors to the variations of nuclear genomes within angiosperms [33,56,57,58,59]. These retroelements are able to replicate using the copy and paste mechanism and, thus, generate new copies of the elements and increase the size of the genome [45]. However, the LTR copies can also be efficiently eliminated from the genome, through both solo LTR formation and accumulation of deletions, which reduces the genome size [54]. Genome size is often treated as an intrinsic property of a species, and intra- and interspecific variations in genome size might reflect different evolutionary processes during speciation [60].

*D. cespitosa* is a variable and widespread species with many subspecies and closely related species (including *D. sukatschewii*) [61], and the genome size of *D. cespitosa* accessions highly depends on their geographical location and habitat [37]. Nevertheless, the average genome size of diploid *D. cespitosa* (1C = 4.38–5.22 pg, Eurasian region) roughly corresponds to that of diploid *D. antarctica* (1C = 4.98–5.31 pg) [36,37,62,63]. These data are consistent with our results showing about the same content of retrotransposons in genomes of the studied *Deschampsia* species, which constituted an essential portion of their repeatomes (41–43%). At the same time, we revealed interspecific differences in content of Ty3-Gypsy and Ty1-Copia and also in genome proportions of SIRE, Angela, non-chromovirus Retand, and chromovirus Tekay. For instance, a ratio of Ty3-Gypsy/Ty1-Copia retrotransposons revealed in genomes of closely related *D. cespitosa* and *D. sukatschewii* differed greatly from that detected in *D. antarctica*. In genomes of *D. cespitosa* and *D. sukatschewii*, the Ty3-Gypsy elements were about 1.5 times more abundant than Ty1-Copia. More content of Ty3-Gypsy retroelements in the genome compared to Ty1-Copia is typical for many taxa of Poaceae. For example, in Avena genomes, Ty3-Gypsy elements were nearly three times more abundant than Ty1-Copia [64]; in genomes of *Lolium* and *Festuca* species, Ty3-Cypsy retrotransposons were four times more abundant compared to Ty1-Copia elements [33]. Moreover, among the studied species, some interspecific variations in the total amount of DNA transposons were detected. The observed interspecific differences might be related to the processes occurred in genomes of these *Deschampsia* species during speciation, which is supported by some previous research. In particular, it was shown that some evolutionary changes in genomes of diploid species of *Melampodium* correlated with differences in the abundance of the SIRE (Ty1-Copia), Athila (Ty3-Gypsy), and CACTA (DNA transposon) lineages [58].

We also found that in *D. antarctica*, the genome proportion of unclassified LTR retroelements significantly exceeded that revealed in the other two *Deschampsia* species, which highlights the need for more research on these TEs in *D. antarctica*. These differences could be related to specific attributes of the *D. antarctica* genome or environmentally induced genetic peculiarities of the studied accessions. Environmentally induced retrotransposon-based genetic diversity was previously described in populations of *D. antarctica* from the Maritime Antarctic [13]. Intense stress might induce rapid changes in the structure, organization, and function of plant genomes especially in populations with low genetic diversity [65], which is typical for *D. antarctica* [66,67,68]. Moreover, in many plant species, which grew under various abiotic and biotic stresses, transcriptional activation of TEs was revealed [69,70,71], and it was regarded as a mechanism responsible for genome plasticity under changing environmental conditions [72].

It was reported for different Poaceae species that satDNAs sequences can vary in a number of features, including nucleotide composition, abundance, and distribution in genomes [73,74]. The comparative analysis of the studied accessions detected interspecific variations in the content of ribosomal DNA, which was notably lower in *D. antarctica* compared to *D. cespitosa* and *D. sukatschewii*. These data are consistent with the different number of satellite chromosomes bearing nucleolar organizer regions (NORs) identified in karyotypes of *D. antarctica* (two pairs) and the other two species (three pairs) [15,16] since it is known that NORs contain tandemly repeated rDNA sequences [75]. Moreover, our results showed that genomes of the studied *Deschampsia* accessions contained substantial portions of satellite DNA sequences, and interspecific variations in their abundance were also revealed. *D. cespitosa* has the highest amount of satellite DNA among the studied species, which is consistent with earlier reported data [34]. Tandem repeats, such as rDNA and other satDNAs, are generally found to be a fast-evolving fraction of the repeatome, showing divergence in both copy number and sequence between closely related species [60]. SatDNAs are known to have a variable length of the repeat unit (monomer) and usually form tandem arrays up to 100 Mb [20,76]. Although they are considered to be non-coding sequences, the satellite monomers mostly exhibit lengths of 160 to 180 bp or 320 to 370 bp though other lengths are also found in plants [77], which correspond to the length of mono- and dinucleosomes [78,79]. The sequences of satellite monomers evolve concertedly via the process of molecular drive; and mutations are homogenized in a genome and become fixed in the populations [80]. The sequence identity inside an array evolves according to the process called ‘concerted evolution’, which results to the maintenance of homogeneity of satDNA monomers within a species during evolution [81]. The abundance of satDNA can vary within the plant genomes even between generations resulting in high polymorphism in the length of satellite arrays [80]. At the same time, some satDNA sequences demonstrate sequence conservation for long evolutionary periods [82]. Since many satellite DNAs exist in a genome, the evolution of species-specific satDNA might be the result of copy number changes within a library of satellite sequences common for a group of species [79,80,82].

The high-throughput DNA sequencing and subsequent genome-wide bioinformatic analysis provide important data on the structural diversity of satDNA [21,83,84]. In the studied accessions of *D. antarctica*, *D. cespitosa*, and *D. sukatschewii*, more satDNA families (20, 27, and 21, correspondingly) were identified by genomic analyses with TAREAN if compared with reported earlier data on South American accessions of *D. antarctica* and *D. cespitosa* (34 satDNAs in total) [34], which indicated a high level of satDNA diversity in *Deschampsia* genomes. Moreover, a relatively large number of the satDNAs were identified in *Deschampsia* genomes compared to several other Poaceae species including *Festuca pratensis* (eight satDNAs), *Agropyron cristatum* (fourteen satDNAs), and *Poa* species (four satDNAs) [31,32,85], which might be related to some features of *Deschampsia* genomes.

Despite satDNAs are considered to be fast-evolving genome fractions, some of them remain preserved for long evolutionary periods and have a highly conserved monomer sequence, which might be related to their interaction with specific proteins necessary for heterochromatin formation and also to their putative regulatory role in gene expression [80,86]. SatDNAs are known to contribute to the essential processes of formation of crucial chromosome structures, e.g., DNA packaging and chromatin condensation [19,79,87,88]. In the present study, three Ds repeats (Ds 56, Ds 83, and Ds 124) showed high sequence similarity with CON1, CON2, and COM2 sequences. CON/COM satDNAs were originally isolated from the *Helictotrichon* genome [22,89] and then revealed in several taxa of the Aveneae/Poeae tribe complex including *Deschampsia* [23,25,90]. In different taxa, the nucleotide sequences in monomers of CON/COM satDNAs demonstrated a high degree of identity, which suggested their ancient origin, though they could change slightly and independently in different species of *Deschampsia* and related genera [22,25,89]. Moreover, BLAST detected regions of local similarity between sequences of several other Ds satDNAs (Ds 52, Ds 81, Ds 88, Ds 138, Ds 142, Ds 166, Ds 179, Ds 182, and Ds 226) and corresponding satDNAs identified in other *Deschampsia* species and/or the species belonged to the related genera, which indicated that those plants might also share a common evolutionary ancestor. Several Ds satDNAs (Ds 65, Ds 144, Ds 158, and Ds 211) had high sequence similarity only with satDNAs of *D. cespitosa* and/or *D. antarctica* confirming their close relationship.

SatDNAs are often associated with heterochromatin regions and are localized in the certain chromosome regions (centromeric, terminal, and/or intercalary), which allow them to be explored with cytogenetic techniques, including FISH. The patterns of chromosomal distribution of satDNAs facilitate the recognition of homologous chromosome pairs and recombination as well as differences between lineages and species [19,20]. High sequence homology of certain satDNAs allowed us to use the oligonucleotide FISH probes, developed based on the most abundant Ds satDNAs, in the comparative karyotype analysis of the studied *Deschampsia* species. However, despite the large number of common repeats, different patterns of chromosomal distribution of these Ds were observed, and depending on the species, localization of most examined Ds satDNAs could be clustered and/or dispersed, which was probably related to different amount and organization of these homologous repeats in genomes of the related species. Large Ds clusters were predominantly localized in the pericentromeric and/or terminal regions of chromosomes of the studied species. Moreover, other patterns of Ds chromosomal distribution were observed including bright clusters combined with dispersed signals or small satDNA clusters in the intercalary chromosome regions, which is typical for plants [19,20,21]. Several Ds satDNAs exhibited only specific clustered localization on chromosomes of all studied species, which allowed us to explore interspecific variations in their distribution on chromosomes. These results were consistent with earlier reported data on patterns of chromosomal distribution of CON/COM and Da satDNAs in several *Deschampsia* species [25,34,35].

According to BLAST, any satDNAs, which would be homologous to Ds 81/Dc 135, were not identified in the *D. antarctica* genome. Moreover, BLAST did not detect any satDNAs homologous to Ds 146 within *Deschampsia* or other taxa. However, the performed FISH-based chromosome mapping of both Ds 81 and Ds 146 revealed bright hybridization signals in karyotypes of all studied *Deschampsia* species. This could be related to some peculiarities of the used sequencing technique, subsequent bioinformatic processing, and also the satDNA abundancy in the genome. Thus, our results demonstrate that the cytogenetic studies can increase the possibilities for satellite DNA analysis as they provide valuable additional data on genomic relationships among related species.

Among the examined Ds tandem repeats, four satDNAs (Ds 52, Ds 81, Ds 65, and Ds 146) demonstrated species-specific patterns of their chromosomal distribution in all studied *Deschampsia* species, which is important for comparative karyotype studies and also analyze the genome differentiation within *Deschampsia*. Specifically, hybridization signals of Ds 52 and also Ds 81 partially overlapped with sites of CON1 satDNA studied previously [25]. Both Ds 65 and Ds 146 demonstrated unique clustered species-specific patterns of chromosomal distribution indicating that they could be used as new promising chromosomal markers for *Deschampsia* species.

SatDNA repeats was shown to represent recombination ‘‘hotspots’’ of genome reorganization, and the occurrence of satDNA in interstitial and telomeric heterochromatin reduces genetic recombination in the adjacent regions [91]. In our study, the comparison of patterns of chromosomal distribution of Ds 65 and Ds 146 made it possible to identify different chromosomal rearrangements in some karyotypes of *D. sukatschewii*, *D. cespitosa*, and *D. antarctica* and detect the breakpoints on chromosomes.

The comparison of patterns of chromosomal distribution of Ds 52, Ds 81, Ds 65, and Ds 146 indicated predominant similarity between karyotypes of *D. sukatschewii* and *D. cespitosa* compared to *D. antarctica*, which was consistent with our previously reported data on other chromosomal markers [16,25]. Notably, the chromosomes bearing 45S and 5S rDNA clusters had the most similar patterns in all three species indicating that structures of these chromosomes were rather conserved. Satellite DNA-based chromosomal markers are particularly useful for chromosome identification, the analysis of chromosome rearrangements, as well as evolution of genomes within Poaceae [24,64,92]. This is especially important for *Deschampsia* due to the lack of effective molecular cytogenetic markers suitable for karyotype analyses within this genus [15]. At the same time, comprehensive genomic studies to assess the variability of satDNA arrays are still required to provide valuable data for investigating the functional and structural features of *Deschampsia* genomes, and also the paths of chromosomal reorganization of genomes during speciation.

## 5. Conclusions

For the first time, the comparative repeatome analyses among valuable subpolar and polar accessions of *D. sukatschewii*, *D. cespitosa*, and *D. antarctica* was performed with the use of the modern effective approach (combining high-throughput DNA sequencing, genome-wide bioinformatic analyses, and FISH-based chromosome mapping of the identified specific satDNAs). Analyses of chromosome patterns of distribution of twelve abundant *D. sukatschewii* satDNAs allowed us to detect four new effective molecular chromosome markers. Due to the shortage of such markers in *Deschampsia*, this is especially important for comparative karyotypic studies within the genus to analyze the changes occurring in their genomes during speciation. For the first time, the unique species karyograms were constructed, which made it possible to compare the localization of these markers on homologous chromosomes of the studied species. Our results confirmed that genomes of the subarctic *D. sukatschewii* and *D. cespitosa* accessions were more closely related if compared with the *D. antarctica* accession according to repeatome composition and patterns of satDNA chromosomal distribution. Our findings demonstrated that cytogenetic studies might expand the possibilities of repeatome analyses as they provide important additional data on genomic relationships within *Deschampsia* as well as increase knowledge on genome organization in these species.

## Figures and Tables

**Figure 1 genes-13-00762-f001:**
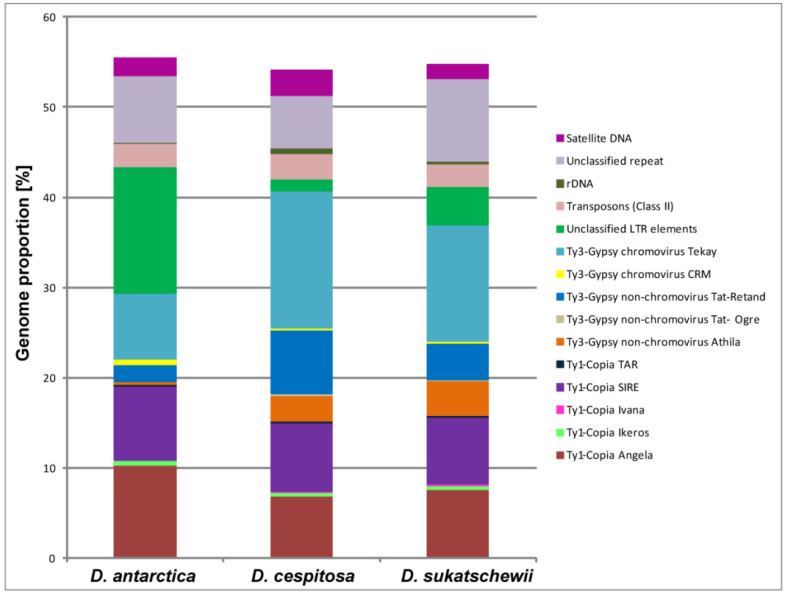
Genome proportion of most abundant repetitive DNA sequences identified in the studied *Deschampsia* species. The genome proportion of individual repeat types was obtained as a ratio of reads specific to individual repeat types to all reads used for clustering analyses by the RepeatExplorer pipelines.

**Figure 2 genes-13-00762-f002:**
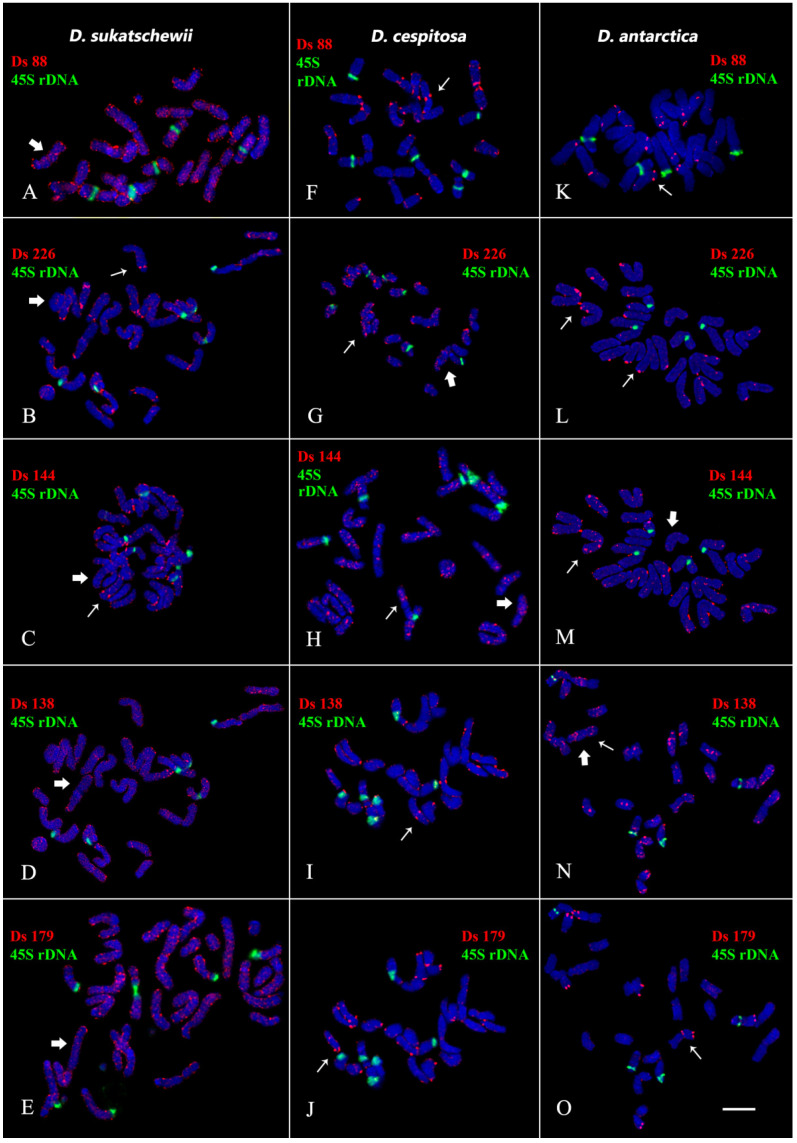
Localization of Ds 88, Ds 138, Ds 144, Ds 179, and Ds 226 satDNA probes on chromosomes of the studied *Deschampsia* species. Merged fluorescent images of *D. sukatschewii*, *D. cespitosa*, and *D. antarctica* after FISH with 45S rDNA (green) and the Ds satDNA probes (red). Chromosome DAPI-staining (grey). (**A**,**B**,**D**,**E**,**G**,**H**,**N**)—mixed clustered and dispersed localization of Ds satDNAs on chromosomes; (**C**,**F**,**I**–**M**,**O**)—clustered localization of Ds satDNAs on chromosomes. Thick and thin arrows indicate dispersed and clustered hybridization signals, respectively. Scale bar—5 μm.

**Figure 3 genes-13-00762-f003:**
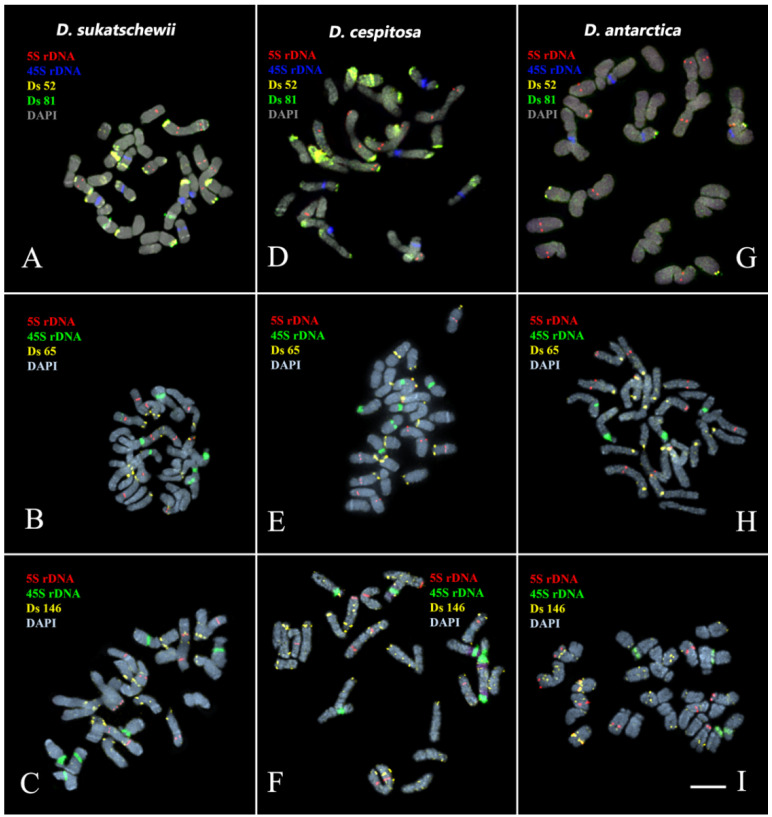
Localization of Ds 52, Ds 65, Ds 81, and Ds 146 satDNA probes on chromosomes of the studied *Deschampsia* species. Merged fluorescent images of *D. sukatschewii*, *D. cespitosa*, and *D. antarctica* after multicolor FISH with 5S rDNA (red), 45S rDNA (blue), Ds 52 (yellow), and Ds 81 (green)—(**A**,**D**,**G**); 5S rDNA (red), 45S rDNA (green), and Ds 65 (yellow)—(**B**,**E**,**H**); and 5S rDNA (red), 45S rDNA (green), and Ds 146 (yellow)—(**C**,**F**,**I**). Chromosome DAPI-staining—grey. Scale bar—5 μm.

**Figure 4 genes-13-00762-f004:**
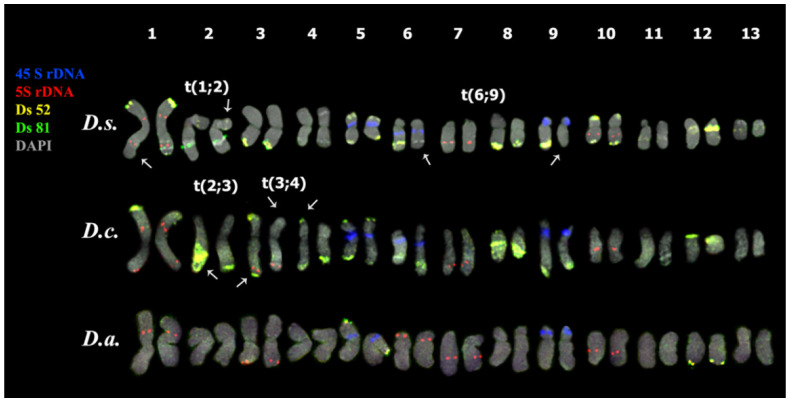
Karyotypes of the studied *Deschampsia* species. Karyograms of *D. sukatschewii* (*D.s.*), *D. cespitosa* (*D.c.*) and *D. antarctica* (*D.a.*) after multicolor FISH with 45S rDNA (blue), 5S rDNA (red), Ds 52 (yellow), and Ds 81 (green) (the same metaphase plates as in Figure 3A,D,G). Chromosome DAPI-staining—grey. Arrows point to chromosomal rearrangements.

**Figure 5 genes-13-00762-f005:**
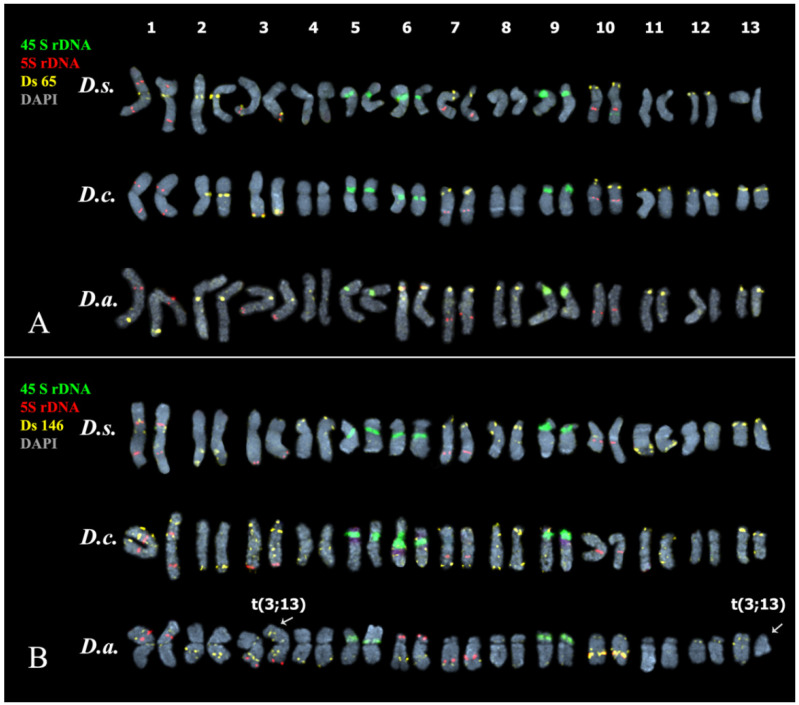
Karyotypes of the studied *Deschampsia* species. Karyograms of *D. sukatschewii* (*D.s.*), *D. cespitosa* (*D.c.*) and *D. antarctica* (*D.a.*) after multicolor FISH with (**A**) 45S rDNA (green), 5S rDNA (red) and Ds 65 (yellow) (the same metaphase plates as in Figure 3B,E,H); and (**B**) 45S rDNA (green), 5S rDNA (red) and Ds 146 (yellow) (the same metaphase plates as in Figure 3C,F,I). Chromosome DAPI-staining—grey. Arrows point to chromosomal rearrangements.

**Table 1 genes-13-00762-t001:** List of the oligonucleotide FISH probes.

Tandem Repeat	Oligo FISH Probe Name/Length, bp	Oligo FISH Probe Sequence
Ds 52	Ds 52_1/20	AATTTGAACCCCTGGACCTC
	Ds 52_2/20	ACCCCTTTTATCCAAATGCC
Ds 56	Ds 56_1/20	ACCAGCTCATTTCGGAACAC
	Ds 56_2/20	AATTCAGGTTCTACGTGCGG
Ds 65	Ds 65_1/21	CTCCAAAACAAAGCTTTGGTG
	Ds 65_1/20	AAGGCTTGTCCATGGAATTG
Ds 81	Ds 81_1/20	GCCTGACACCCTGACTTAGC
	Ds 81_2/20	GAAAAGATGCACTGATCGCA
Ds 83	Ds 83_1/20	GCCAGAAGTATCCCAAACGA
	Ds 83_2/20	TAGTGTGTTATGGCCCACGA
Ds 88	Ds 88 _1/23	CGGTTTCGAAGGCGTTAGAAAGA
	Ds 88 _2/20	ACTCGAAATTCGATGGAACG
Ds 124	Ds 124_1/20	TGCAAAATTTCTTGACACCG
	Ds 124_2/20	GCGTGAAATTCCCACAGATT
Ds 138	Ds 138_1/20	GTCTACCCCTTTGACCGGAT
	Ds 138_2/20	CCAATGAACGTTTTCCTTCC
Ds 144	Ds 144_1/20	GGGGGTAGCTCAATGGAACT
	Ds 144_2/20	TTATGTTCATTTGTGTTTGT
Ds 146	Ds 146_1/20	ATACCACCTTGTGAAAAGTA
	Ds 146_2/20	TCCCTTTCCTCATTGGATCA
Ds 179	Ds 179_1/23	ATGGCACATGATGAAACGCGTTT
	Ds 179_2/20	TTTAATACGGGACTGGGCTG
Ds 226	Ds 226_1/20	AGCATGGAAAACCAAGTTGG

**Table 2 genes-13-00762-t002:** Proportion of major repetitive DNA sequences identified in genomes of the studied *Deschampsia* species.

Repeat Name	Genome Proportion (%)		
	*D. antarctica*	*D. cespitosa*	*D. sukatschewii*
**Retrotransposons (Class I)**	**43.41**	**42.13**	**41.21**
**Ty1-Copia**	**19.21**	**15.27**	**15.77**
Unclassified Ty1-Copia elements	-	0.17	0.01
Ale	-	0.02	0.02
Angela	10.28	6.84	7.49
Ikeros	0.51	0.34	0.47
Ivana	-	0.18	0.18
SIRE	8.25	7.54	7.38
TAR	0.17	0.18	0.22
**Ty3-Gypsy**	**10.13**	**25.53**	**21.16**
non-chromovirus Athila	0.27	2.85	3.83
non-chromovirus Tat- Ogre	-	0.22	0.13
non-chromovirus Tat-Retand	1.94	7.02	4.09
chromovirus CRM	0.56	0.23	0.22
chromovirus Tekay	7.36	15.21	12.89
**LINE**	**0.03**	**0.02**	**0.01**
**Unclassified LTR elements**	**14.03**	**1.31**	**4.27**
**Transposons (Class II)**	**2.58**	**2.89**	**2.48**
Cacta	2.44	2.84	2.34
MuDR_Mutator	0.13	0.01	0.12
PIF_Harbinger	0.01	0.04	0.02
**Ribosomal DNA**	**0.06**	**0.6**	**0.29**
**Unclassified repeats**	**7.45**	**5.86**	**9.18**
**Satellite DNA**	**2.07**	**2.85**	**1.61**
**Organelle**	**4.81**	**1.04**	**2.73**
**Repetitive DNA**	**60.37**	**55.37**	**57.5**
**Putative satellites**	**13 high confident** **7 low confident**	**18 high confident** **9 low confident**	**12 high confident** **9 low confident**

**Table 3 genes-13-00762-t003:** Comparison of satDNAs identified in *Deschampsia sukatschewii* with our results on *D. cespitosa and D. antarctica* and also available data.

SatDNA/Genome Proportion, %/Repeat Length, bp (Our Data) *			BLAST Homology (Available NCBI Data)
*D. sukatschewii*	*D. cespitosa*	*D. antarctica*	
Ds 52/0.33/184	Dc 87/0.23/184	not found	*Poa pratensis* clone PpTR-3 microsatellite sequence KY618841.1, *Poa pratensis* clone PpTR-2 microsatellite sequence KY618840.1, 78% of identity with Ds 52.
Ds 56/0.29/366	Dc 38/0.48/366	Da 272/0.025/366	*D. cespitosa* satellite D12 sequence MT548102.1, *D. antarctica* clone 1 satellite D12 sequence MT548072.1, 99% of identity with Ds 56, *H. convolutum* satellite DNA (ID: pCON1_3).
Ds 65/0.22/314	Dc 89/0.23/314	not found	*D. cespitosa* satellite D31 sequence MT548119.1, *D. antarctica* satellite D31 sequence MT548089.1, 97–98% of identity with Ds 65.
Ds 81/0.14/369	Dc 135/0.078/369	not found	*Leymus triticoides* clone Lt1-4 satellite sequence EU629350.1, 83% of identity with Ds 81.
Ds 83/0.13/355	Dc 125/0.1/355	Da 197/0.076/355	*D. antarctica* clone 2 satellite D17 sequence MT548144.1, *D. cespitosa* satellite D17 sequence MT548106.1, 70–98% of identity with Ds 83, *H. compressum* satellite DNA (ID pCOM2_4) Z68786.1.
Ds 88/0.11/379	Dc 17/0.69 380 (75% of identity with Ds 88)/; Dc 77/0.31/379	DA 238/0.042/379	*D. cespitosa* satellite D10 sequence MT548100.1, *D. antarctica* satellite D4 sequence MT548064.1, 96–99% of identity with Ds 88, *Secale cereale* clone BAC 114I10 satellite pSc200 sequence KT724946.1, 83% of identity with Ds 88.
Ds 124/0.046/569	Dc 234/0.016/563	Da 351/0.013/563	*D. antarctica* satellite D13 sequence MT548073.1, *D. cespitosa* satellite D13 sequence MT548103.1, 99–100% of identity with Ds 124, *H. convolutum* satellite DNA (ID pCON2_2).
Ds 138/0.036/158	Dc 177/0.031/158	Da 154/0.11/158	*D. cespitosa* satellite D5 sequence MT548095.1, *D. antarctica* satellite D6 sequence MT548066.1, 100% of identity with Ds 138, *D. antarctica* clone 1 satellite D5 sequence MT548133.1, 94% identity with Ds 138, *D. cespitosa* satellite D29 sequence MT548117.1, *D. antarctica* satellite D29 sequence MT548087.1, *D. antarctica* satellite D20 sequence MT548081.1, *D. cespitosa* satellite D20 sequence MT548110.1, 73–77% of identity with Ds 138, *Festuca pratensis* satellite TR7 sequence.
Ds 142/0.035/658	Dc 163/0.041/658	Da 107/0.18/658	*D. antarctica* satellite D5 sequence MT548065.1, *D. antarctica* clone 2 satellite D4 sequence MT548131.1, 99% of identity with Ds 142, *Triticum aestivum* cultivar Chinese Spring clone BAC 36I14, complete sequence.
Ds 144/0.034/352	Dc 146/0.063/352	Da 322/0.013 /342 (70% of identity with Ds 144)	*D. antarctica* clone 3 satellite D1 sequence MT548124.1, *D. cespitosa* satellite D7 sequence MT548097.1, both 72–73% of identity with Ds 144.
Ds 146/0.033/344	not found	not found	not detected.
Ds 158/0.03/350	Dc 238/0.016/358	Da 116/0.17/351	*D. antarctica* satellite D16 sequence MT548076.1, *D. cespitosa* satellite D16 sequence MT548105.1, 98–94% of identity with Ds 158, *D. cespitosa* satellite D22 sequence MT548111.1, *D. antarctica* satellite D22 sequence MT548083.1, 67% of identity with Ds 158.
Ds 166/0.027/174	Dc 302/0.01/174	not found	*D. antarctica* satellite D25 sequence MT548086.1, *D. cespitosa* satellite D25 sequence MT548113.1, 98–99% of identity with Ds 166.
Ds 179/0.023/318	Dc 261/0.013/318	not found	*D. antarctica* satellite D3 sequence MT548063.1, *D. cespitosa* satellite D3 sequence MT548094.1, 91–89% of identity with Ds 179, *Setaria viridis* cultivar ME034v chromosome 1.
Ds 182/0.022/343	Dc 211/0.02/343	Da 225/0.051/341	*D. cespitosa* satellite D33 sequence MT548121.1, *D. antarctica* satellite D33 sequence MT548091.1, 70% of identity with Ds 182, *D. cespitosa* satellite D20 sequence MT548110.1, *D. antarctica* satellite D20 sequence MT548081.1, 74% of identity with Ds 182, *D. cespitosa* satellite D5 sequence MT548095.1, *D. antarctica* satellite D6 sequence MT548066.1, 70% of identity with Ds 182, *H. pratense* satellite DNA (ID pPRA1_2).
Ds 211/0.015/171	Dc 174/0.034/171	Da 204/0.067/171	*D. cespitosa* satellite D6 sequence MT548096.1, *D. antarctica* satellite D7 sequence MT548067.1, *D. antarctica* clone 1 satellite D6 sequence MT548134.1, 98% of identity with Ds 211.
Ds 226/0.014/345	Dc 106/0.16/ 352 (67% of identity with Ds 226)	Da 97/0.21 /342; Da 129/0.14/343	*D. cespitosa* satellite D1 sequence MT548092.1, *D. antarctica* satellite D1 sequence MT548061.1, 75% of identity with Ds 226, *D. cespitosa* satellite D20 sequence MT548110.1, *D. cespitosa* satellite D5 sequence MT548095.1, *D. antarctica* satellite D6 sequence MT548066.1, 70% of identity with Ds 226, *Festuca pratensis* satellite TR4 sequence.

* By default, the identity of Dc and Da satDNAs with the corresponding Ds satDNA is 98–100%.

## Data Availability

The data presented in this study are contained within the article and Appendix A. The datasets of *D. cespitosa* (SAMN26938767) and *D. sukatschewii* (SAMN26938768), generated during this study, can be found here: https://www.ncbi.nlm.nih.gov/bioproject/PRJNA819861 (accessed on 25 March 2022), BioProject accession number PRJNA819861.

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
