# Peer review of "Repeatome Analyses and Satellite DNA Chromosome Patterns in Deschampsia sukatschewii, D. cespitosa, and D. antarctica (Poaceae)"

_genes, 2022, doi:10.3390/genes13050762_

Round 1

Reviewer 1 Report

Dear Author,

The manuscript is on Repeatome Analyses and Salellite DNA Chromosome Patterns in Deschampsia sukatschewii, D. cespitosa and D. antarctica  (Poaceae).  I have gone trhrough the manuscript and I found some scientific mistake which sould be corrected. 

  • Replace “Salellite” with “Satellite” in the title.
  • Write the abbreviation of “TAREAN” in the first place you mention (delete from Line 87).
  • Replace “fluorimeter” with “fluorometer” (Line 107).
  • Clarify this part of the sentence “genome DNA low-coverage sequencing”. Did you mean “whole genome sequencing with low coverage?”? (Line 109). Please emphasize that
  • Raw sequencing data for cespitosa and D. sukatschewii were not found in the given links. Please provide proper links. Also provide NCBI BioProjects numbers (Line 114-115).
  • Write all subheadings in a same format (Line 117).
  • Please indicate the citation for RepeatExplorer/TAREAN pipelines specifically (Line 123).
  • Indicate the quality filtering parameters (what is the parameters ?, quality score etc.)? (Line 123-124).
  • Replace “Cromosome” to “Chromosome” (Line 140).
  • Did you allow them to dry out in a certain time period dehydration and air dry process? or what (Line 147, 161, 167).
  • Please indicate the magnification power (What X) (Line 173).
  • Check the % signs in the article. Delete extra spaces between number and % sign.
  • Write all table captions in same format.
  • Use passive sentence instead of active (Line 12, 149-150, 152-153, 405, 422, 452, 458). Do not use word or sentence indicate possession
  • Please indicate the x and y axes in Figure 1.
  • Replace “Figure1” as “Figure 1” in Line 200.
  • Please write “Blast” in capital letters (Line 210).
  • Write species names italic in Figure 2 and Figure 3.
  • The manuscript was uploaded into the Ithentice to detect plagiarism and there are too many sentence similarities from “ncbi.nlm.nih.gov” database. Please rewrite the sentences from Line 228 to Line 240.
  • Replace “LTR-retrotransposons” to “LTR retrotransposons” in Line 277.
  • The manuscript was uploaded into the Ithentice to detect plagiarism and there are too many similarities with “Durdica Ugarkovic. "Functional elements residing within satellite DNAs", EMBO reports, 2005”. Please rewrite the sentences from Line 373 to Line 375.
  • Indicate the possible reasons of obtaining relatively large number of satDNA families compared to earlier studies. (Line 364-Line 370).
  • It was pointed out that you performed for the first time the comparative repeatome analyses of sukatschewii, D. cespitosa and D. antarctica using RepeatExplorer/TAREAN pipelines. Are there any other research using RepeatExplorer/TAREAN pipelines in Poaceae species? If any, compare the results of these studies with your results.

Reviewer 2 Report

Please see the following comments for the editor and authors.

The major research objective of this study is repeatome analysis using low coverage (0.147 – 0.350x of the coverage) of the NGS next genome Deschampsia genome and FISH chromosomal mapping of 45S rDNA, 5S rDNA and satDNAs of D. sukatschewii, D. cespitosa and D. Antarctica. This allowed us to construct the species karyograms and new molecular chromosome markers Ds 52, Ds 81, Ds 65 and Ds 146 to identify 13 chromosomes.

The experimental procedure and data are simple, providing clear results and easy to understand.

 [Comments for the revisions]

1) Table 4 is not informative for subscribers. I recommend that it be included as supplemental data. The literature description of the characteristics of DS in Table 4 is not easy to understand.

2) Instead of present Table 4, the authors had better to show the quantitative data of the chromosome length and arm ratio of 13 pairs chromosomes and the Ds markers localization positions.

3) P.16, L. 445: Delete “,“ in [15,]

Reviewer 3 Report

The study entitled "Repeatome Analyses and Salellite DNA Chromosome Patterns in Deschampsia sukatschewii, D. cespitosa and D. antarctica (Poaceae) " has been conducted throughly by the authors. Overall, this study represents an in-depth Repeatome Analyses and Salellite DNA Chromosome Patterns in Deschampsia sukatschewii, D. cespitosa and D. antarctica (Poaceae). The paper is well organized, easy to follow, and of potential importance both in terms of scientific discovery but also commercially. Authors have done a good job of covering the literature.

Line 103: Genomic DNAs of D. sukatschewii and D. cespitosa were isolated from young leaves. Please write DNA instead of DNAs. 

Line 140: Cromosome Spread Preparation, please replace cromosome by chromosome.

Round 2

Reviewer 1 Report

Dear Author, The current version of the manuscript has already been revised and corrected all scientific and gramatical shortcomings raised by previous round of the review. 

I think the current version of the manuscript deserves to be accepted for publication. 

All the Best